# Advancing the scale of synthetic biology via cross-species transfer of cellular functions enabled by iModulon engraftment

Donghui Choe [1], Connor A. Olson[1], Richard Szubin[1], Hannah Yang[1], Jaemin Sung[1], Adam M. Feist [1,2] & Bernhard O. Palsson [1,2] ✉

Machine learning applied to large compendia of transcriptomic data has enabled the decomposition of bacterial transcriptomes to identify independently modulated sets of genes, such iModulons represent specific cellular functions. The identification of iModulons enables accurate identification of genes necessary and sufficient for cross-species transfer of cellular functions. We demonstrate cross-species transfer of: 1) the biotransformation of vanillate to protocatechuate, 2) a malonate catabolic pathway, 3) a catabolic pathway for 2,3-butanediol, and 4) an antimicrobial resistance to ampicillin found in multiple *Pseudomonas species* to *Escherichia coli*. iModulon-based engineering is a transformative strategy as it includes all genes comprising the transferred cellular function, including genes without functional annotation. Adaptive laboratory evolution was deployed to optimize the cellular function transferred, revealing mutations in the host. Combining big data analytics and laboratory evolution thus enhances the level of understanding of systems biology, and synthetic biology for strain design and development.

Synthetic biology aims to engineer new, or modify existing, cellular functions by designing, editing, and assembling the underlying genetic material. To achieve this goal, knowledge-based designs have been traditionally used that identify genes of known molecular functions to piece together the targeted cellular function. However, identifying all the necessary genetic components is not trivial and requires large-scale analysis such as genome-wide knockout studies and Tn-Seq[1,2]. This approach also calls for serial steps of genetic transformation with separate testing of each individual gene of the targeted function[3]. For example, the development of the biosynthetic pathway of the antimalarial drug artemisinin demanded extensive research efforts and engineering[4–6]. To circumvent the difficulties, computational tools have been developed to supplement experimental procedures and aid design[7–9]. More importantly, recent advances in new data analytics applied to large transcriptomic datasets has enabled an alternative approach. Using Independent Component Analysis (ICA), we can now identify sets of independently

modulated genes (called iModulons) that constitute a particular cellular function[10].

This advancement opens up the possibility to identify targeted functions in a particular strain and transfer them into alternate hosts. Successful cross-species transfer of desired functions would thus require: (i) the identification of the full genetic basis for the trait, (ii) the use of recombinant or DNA synthesis methods to capture these genes into a plasmid, (iii) the transfer of the plasmid into the target host, and (iv) making any needed changes to the new host that are critical to accommodate the transferred function. All these capabilities now exist, with (ii) and (iii) representing known approaches, while (i) and (iv) require novel transcriptomic analysis and the use of automated adaptive laboratory evolution.

ICA can be applied to find source regulatory signals in bacterial transcriptomes[10–13]. iModulons are fundamental units of bacterial transcriptomes that have been found to represent the genetic basis for various cellular functions and are associated with particular

[1]Department of Bioengineering, University of California San Diego, La Jolla, CA 92093, USA. [2]Novo Nordisk Foundation Center for Biosustainability, Technical University of Denmark, Copenhagen, Denmark. ✉e-mail: bpalsson@ucsd.edu

transcriptional regulator(s)[14–16]. Many identified iModulons include genes that are distantly located on the genome, genes of unknown functions, or contain accessory genes that augment the targeted cellular function. iModulons thus represent an advanced scale of synthetic biology to transfer naturally evolved traits across species.

Here, we use iModulons to create cellular functions in a new host. We show that transferring iModulons is superior to using operons or single genes identified by genome annotation algorithms and rational design approaches[9,17,18]. In some cases, the transferred function may not work optimally in the new host. In such cases, we can use adaptive laboratory evolution (ALE) to enable the host to optimally use the new function under selection pressure. We thus demonstrate that cross-species iModulon transfer is a versatile tool for synthetic biology.

## Results

### Cross-species transfer of *Pseudomonas* iModulons into *E. coli*
To initiate the project and prior to implementing cross-species iModulon transfer, we refactored a known cellular function within the original host as a proof of concept. Successful homologous refactoring and complementation of *E. coli*'s branched-chain amino acid (BCAA) metabolism was achieved (Supplementary Note, section 1 and Supplementary Fig. 1) to demonstrate identification, reconstruction, and transfer of genetic constituent of a biological function based on iModulon (i, ii, and iii). This motivated us to investigate the potential for transferring biological functions across species. Among the available species with iModulon structures in iModulonDB[12], *Pseudomonas* is well-known for its versatile metabolism to degrade and utilize diverse compounds, including aromatics[19–21]. First, we chose to reconstruct and transfer a simple bioconversion process from *Pseudomonas putida*[15] to *E. coli* in order to examine iModulon's capability to rapidly identify genes associated with specific functions.

The VanR iModulon that is responsible for vanillate (VA) transport and conversion into protocatechuate (PCA) was chosen for our first cross-species iModulon transfer (Fig. 1A). It comprises three genes with annotated functions, *vanA*, *vanB*, *vanK*, and predicted porin-like *galP*-IV (Fig. 1B) in two converging operons (Fig. 1C). Notably, the iModulon exactly matches with the genes for the vanillate transport and metabolism[22,23]. Four genes, *vanA*, *vanB*, *galP*, and *vanK* are functionally annotated to encode for vanillate O-demethylase oxidoreductase complex, outer-membrane porin, and a major facilitator superfamily transporter, respectively[22]. Although the function of the outer membrane OprD-domain containing *galP*-IV has never been addressed, it is hypothesized that it facilitates the diffusion of the ligand through the outer membrane[23,24]. Since the mechanism of VanR regulation has not been established, the four genes constituting the VanR iModulon were cloned and heterologously expressed under the control of IPTG-inducible Trc promoter on a plasmid, pVanR_iM (Fig. 1D). When refactoring iModulons for heterologous expression, we tried to preserve native genetic arrangement, for VanR and following iModulons if possible, to ensure optimal expression levels of the gene members as demonstrated elsewhere[25,26].

*E. coli* carrying pVanR_iM converted VA into PCA up to 15.34 mg/l passively diffused to the supernatant[27,28] during 48 h of fermentation in M9 glucose (4 g/l) medium supplemented with 100 mg/l VA, while the negative control carrying empty plasmid did not metabolize any VA (Fig. 1E). This first cross-species iModulon transplantation illustrates the rapid identification of enzymes required for biotransformation by ICA. Furthermore, iModulon engraftment provided a rapid way to biochemically verify a predicted pathway in a heterologous host.

### Auxiliary genes may be needed for optimal function of cross-species transferred iModulons
Next, we chose to transfer an ampicillin resistance function of *Pseudomonas aeruginosa* to *E. coli*. *P. aeruginosa* displays beta-lactam resistance with endogenous beta-lactamase, AmpC, and has an iModulon involved in the inducible ampicillin resistance[16]. Activity levels of the AmpC iModulon are highly induced against beta-lactam challenge, but not under other antibiotic treatments (Supplementary Fig. 2). In the previous iModulon engraftment examples, genes comprising an iModulon matched with the predicted genes necessary for building the desired function. However, identifying all the genes necessary to build a biological function may not be trivial, given previous characterization efforts. Many iModulons contain genes whose functions are unknown or are seemingly unrelated to the overall function being transferred.

The AmpC iModulon comprises class C beta-lactamase encoded by the *ampC* gene[29] that serves as a core for the functionality and six lesser characterized auxiliary genes, *carO* (PA0320), *creD* (PA0465), PA0466, PA0467, PA4111, and PA4112 (Fig. 2A). The seven iModulon genes are distributed across three genomic loci separated by over 4 Mb. *P. aeruginosa* readily becomes resistant to ampicillin by transcriptional activation of *ampC*[30]. However, it is not known if the resistance trait is carried by this single gene. To examine if this resistance function is transferable across species, the constituent genes were refactored into a single operon (Fig. 2B). In addition, we constructed a plasmid that contained beta-lactamase alone to address any involvement of auxiliary factors in the function.

Ampicillin disc diffusion assay revealed that *E. coli* carrying the AmpC iModulon or *ampC* gene were resistant to ampicillin, while *E. coli* carrying empty plasmid were not (Supplementary Fig. 3). The source of AmpC iModulon, *P. aeruginosa*, showed ampicillin resistance with the minimum inhibitory concentration (MIC) of 2048 μg/ml (Fig. 2C). The MIC of ampicillin for laboratory *E. coli* strain MG1655 with empty plasmid was 16 μg/ml, which is comparable to previous reports[31,32] (Fig. 2C, D). *E. coli* strain with the *P. aeruginosa* beta-lactamase showed a dramatic increase in ampicillin resistance with an MIC of 1024 μg/ml, while it was lower than that of the original host (Fig. 2D). Strikingly, *E. coli* harboring the entire AmpC iModulon, six auxiliary genes in addition to *ampC*, had an MIC of 4096 μg/ml, which was four times higher than that with *ampC* alone (Fig. 2D).

Although little is known about the molecular function of auxiliary genes, they were required to completely replicate the ampicillin resistance characteristics of *P. aeruginosa*. Previous reports have shown a decrease in beta-lactam resistance of the inner membrane protein *creD* knockout mutant of *P. aeruginosa*[33] and growth enhancement of *E. coli* by endogenous *creD* overproduction (shares 37.4% sequence identity; BLOSUM62)[34]. Although the function of CreD is still elusive, reports indicate its relevance in biofilm development in *P. aeruginosa*[35] and envelope integrity in *Stenotrophomonas maltophilia*[36]. Additionally, calcium-regulated oligonucleotide/oligosaccharide binding (OB)-fold protein CarO has been reported to be related to susceptibility to various stresses in bacteria[37]. Also, it shares similarity with *Salmonella enterica* stress-related protein VisP (38% sequence identity), which binds to peptidoglycan and inhibits the lipid A modifying enzyme LpxO[38]. Since lipid A is an anchor of lipopolysaccharide to the outer membrane and affects the properties of the outer membrane, expression of *carO* might be beneficial for cells to maintain structural integrity under cell wall deficient conditions induced by beta-lactam[39].

Engrafting *Pseudomonas* iModulons to *E. coli* highlighted critical properties of iModulon gene membership. Harnessing only core genes for transferred cellular function may not be sufficient, as auxiliary genes may be needed to reconstruct an optimal function. Full iModulon gene membership helps to recreate the targeted cellular function, even without a complete understanding of the molecular function of all the genes involved.

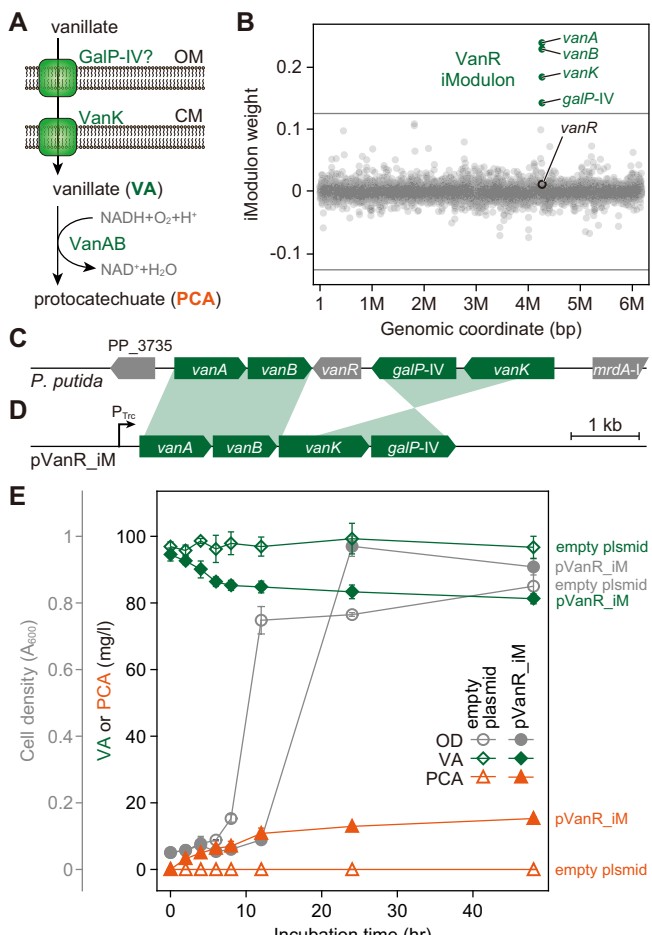

**Fig. 1 | Cross-species transfer of the *Pseudomonas putida* VanR iModulon into *E. coli*. A** Vanillate transport and conversion in *P. putida*. OM outer membrane. CM cytoplasmic membrane. **B** iModulon weights of genes in *P. putida*. Four genes (green circles) with high weighting constitute the VanR iModulon. Gray lines indicate thresholds for determining iModulon membership. Gray circles identify genes not in the iModulon. **C** Graphical representation of *vanR* locus on the *P. putida* chromosome. **D** The VanR iModulon was refactored in a single operon under the control of *trc* promoter (P$_{Trc}$), resulting in the pVanR_iM plasmid. Shades show genetic rearrangement for cloning purposes. **E** Vanillate (VA) conversion of *E. coli* carrying empty or pVanR_iM plasmid into protocatechuate (PCA). Gray circles, green diamonds, and orange triangles indicate cell density, VA, and PCA levels of the culture, respectively. Measurements from *E. coli* carrying empty or pVanR_iM plasmid are represented by hollow or filled symbols, respectively. Data were presented as mean values ± SD. Error bars indicate the SD of three replicate cultures. Source data are provided as a Source Data file.

## Complete iModulon gene membership is needed for successful cross-species transfer

As illustrated by the AmpC case, we further investigated the iModulon-based transfer of cellular traits and compared it to the alternative conventional methods. The 2,3-butanediol (2,3-BDO) iModulon was chosen to examine the role of iModulon genes of unknown functions. 2,3-BDO is a byproduct of bacterial fermentation processes that can be produced by a variety of microorganisms, including *Pseudomonas* species[40–42]. In *Pseudomonas*, 2,3-BDO can serve as a carbon and energy source and is degraded by enzymes in the 2,3-BDO catabolic pathway[42]. This catabolic pathway involves the conversion of 2,3-BDO into acetoin, which is further converted into acetaldehyde and acetyl-CoA by butanediol dehydrogenase and acetoin dehydrogenase, respectively (Fig. 3A).

We transferred the 2,3-BDO iModulon of *P. putida* (called the AcoR iModulon[15]) to *E. coli*. The AcoR iModulon comprises *acoABC*

(encoding acetoin dehydrogenase complex), *bdhA* (encoding 2,3-BDO dehydrogenase), and a gene *acoX* (Fig. 3B). AcoX encodes for a protein of unknown function and co-exists with acetoin-utilizing genes in various bacteria[41,43]. Operon prediction also suggests that the transcriptional unit contains *acoX* and two other hypothetical proteins (PP_0550 and PP_0551) in addition to characterized metabolic enzymes, *acoABC-bdhA* (Fig. 3C)[18,44].

To examine which genes are required for recreating the 2,3-BDO catabolic pathway, we built three different plasmid based on (1) operonic structure (Op353; *acoXABC-bdhA*-PP_0551-PP_0550), (2) iModulon structure (*acoXABC-bdhA*), and (3) four genes encoding enzymes predicted to be sufficient for converting 2,3-BDO into acetaldehyde and acetyl-CoA based on current gene annotations (pathway; *acoABC-bdhA*) (Fig. 3C). 2,3-BDO dehydrogenase activities of the source organism and *E. coli* strains carrying the three plasmids individually were examined during 96 h of batch cultivation in LB medium supplemented with 2 g/l of 2,3-BDO. The original strain, *P. putida* KT2440, showed 2,3-BDO utilization with a negligible level of acetoin (Fig. 3D). The negative control, *E. coli* MG1655 carrying an empty plasmid converted 0.77 g/l of 2,3-BDO into acetoin, possibly due to endogenous promiscuous alcohol dehydrogenase activity (Fig. 3E). On the other hand, the plasmids based on the pathway, operonic structure, and iModulon showed higher conversion of 2,3-BDO with amounts of 1.36, 1.75, and 1.96 g/l, respectively (Fig. 3E).

Interestingly, the strains showed varying levels of acetoin dehydrogenase activity. First, all the 2,3-BDO consumed by the negative control resulted in roughly the equimolar amount of acetoin; not surprising since there is no acetoin dehydrogenase introduced. The strain carrying the functional gene annotation-based pathway plasmid did not further convert acetoin into downstream products, even though it contained genes encoding for the acetoin dehydrogenase complex. Second, strains with the full operon or AcoR iModulon not only consumed more than 1.7 g/l of 2,3-BDO, but there was only a small amount of acetoin left in the medium, indicating conversion of acetoin by acetoin dehydrogenase. The difference between annotation-based and iModulon-based plasmid is the presence of *acoX* (Fig. 3C), a gene encoding a predicted small molecule kinase that has been reported to have no acetoin, NAD, or pyruvate kinase activity[45]. However, *acoX* was critical for acetoin dehydrogenase activity.

Although the *acoX* product has no known function in acetoin metabolism, it is conserved and colocalizes on the genome with the acetoin dehydrogenase in several acetoin-utilizing bacteria from multiple phyla, such as *P. aeruginosa* (76% sequence identity) and *Clostridium magnum* (32% sequence identity)[42]. However, there is no significant match of AcoX from the BLASTP search on other acetoin-utilizing bacteria such as *Bacillus subtilis*, *Klebsiella pneumoniae*, and *Pelobacter carbinolicus*. Therefore, the requirement of AcoX in acetoin metabolism is species-specific and could not be determined by analyzing the genome sequence context.

When the iModulon and operonic constructs were compared, the iModulon construct performed better than the operonic construct for 2,3-BDO degradation (Fig. 3E). Two additional genes in the operonic construct encode the predicted membrane occupation and recognition nexus (MORN) domain-containing peptidase and a NAD(P)-binding oxidoreductase, whose relation with 2,3-BDO metabolism is unknown. These two genes were irrelevant for function. Instead, expression of the hypothetical proteins reduced 2,3-BDO degradation, possibly by imposing an unnecessary transcriptional burden on the cell. The iModulon gene membership provided information on the necessary genes to support a 2,3-BDO catabolic process that would not have been found using only functional gene annotation. This example illustrates the unique advantages of using the iModulon structure for cross-species transfer of the full genetic basis for a desired integrated function.

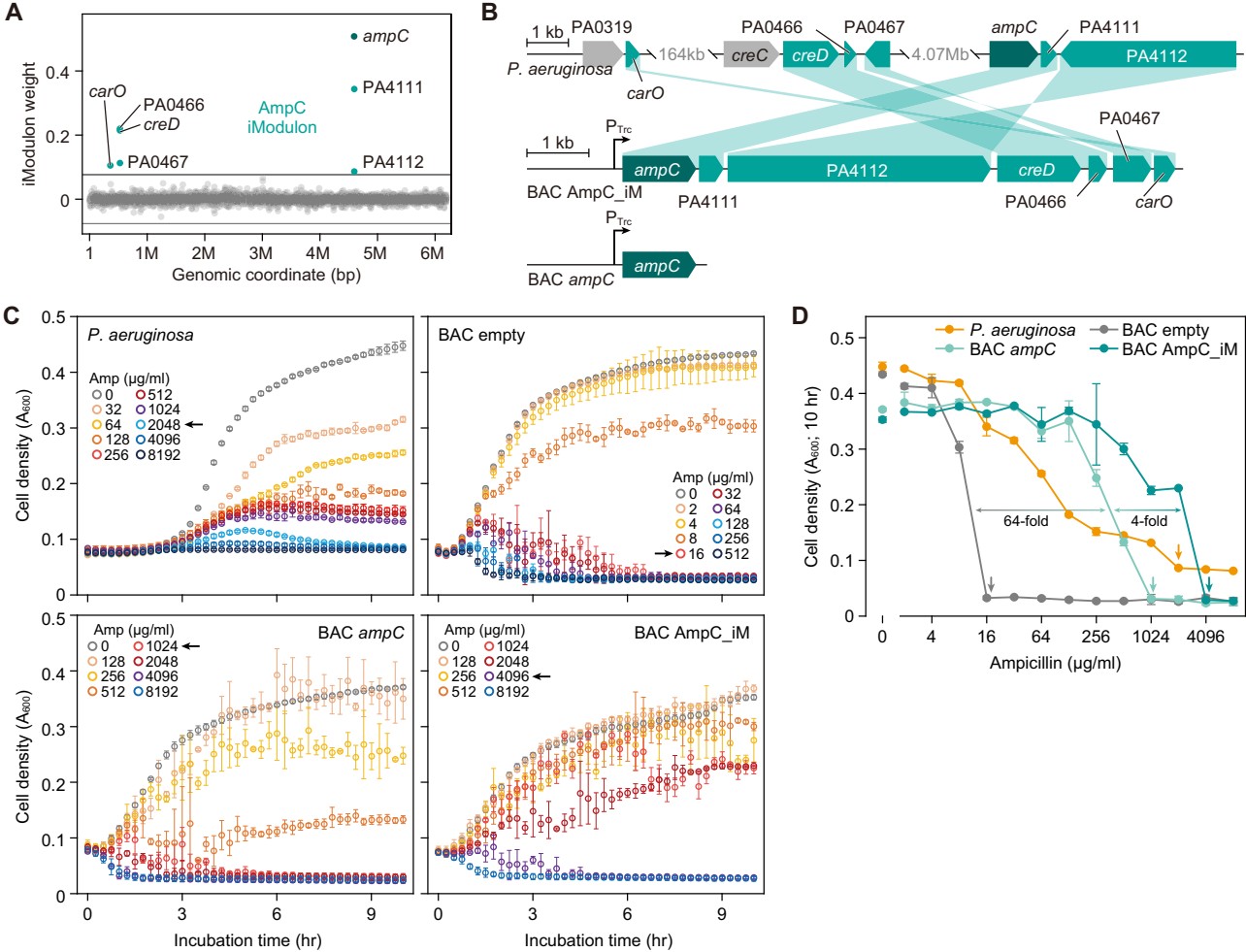

**Fig. 2 | *E. coli* carrying the *Pseudomonas aeruginosa* AmpC iModulon confers better ampicillin resistance than cells expressing beta-lactamase alone.**
**A** iModulon weights of genes in *P. aeruginosa*. Seven genes constitute the AmpC iModulon (blue circles). Gray lines indicate thresholds for determining iModulon membership. Gray circles identify genes not in the iModulon. **B** Refactoring the *P. aeruginosa* AmpC iModulon on bacterial artificial chromosome (BAC). Genes are expressed with the *trc* promoter ($P_{Trc}$). Shades show genetic rearrangement for cloning purposes. **C** Dose-kill curves of *P. aeruginosa* and *E. coli* carrying empty BAC, BAC_*ampC*, or BAC_AmpC_iM. Data were presented as mean values ± SD. Error bars indicate the SD of biological replicates ($n = 3$). Note that the range of ampicillin concentration (Amp) is different, due to the huge difference in ampicillin tolerance. **D** Cell density of cultures treated with different ampicillin concentrations after 10 h of incubation. Data for *P. aeruginosa* and *E. coli* carrying empty BAC, BAC_*ampC*, or BAC_AmpC_iM are in orange, gray, light blue, and blue, respectively. Arrows indicate the minimum inhibitory concentration (MIC). Data were presented as mean values ± SD. Error bars indicate the SD of biological replicates ($n = 3$). Source data are provided as a Source Data file.

## ALE optimizes the functionality of catabolic iModulons

Lastly, we chose the MdcR iModulon from *P. aeruginosa*[16] to transfer into *E. coli* that, again, comprises genes identical to a reported set for malonate transport and utilization. The MdcR iModulon comprises seven subunits of malonate decarboxylase complex[21] and two putative membrane proteins, MadL-MadM (Fig. 4A and Supplementary Fig. 4). Although the function of these membrane proteins have not been elucidated in *P. aeruginosa*, MadL and MadM have 71 and 81% of sequence homology to malonate transporters in *Malonomonas rubra*[46], respectively, suggesting a potential malonate uptake function. These genes are encoded in a single operon on the *P. aeruginosa* genome, and thus the entire operon was subjected to cross-species transfer.

The operon was cloned and heterologously expressed under the control of a Trc promoter on a plasmid, named pMdcR_iM (Fig. 4B). Malonate is a non-native nutrient for *E. coli*, thus it is expected that a strain with the pMdcR_iM alone would then enable growth in M9 malonate medium as the breakdown product of the pathway, acetate, can support growth[47]. We experimented with varying levels of

expression using different concentrations of the inducer (IPTG) to activate the MdcR iModulon. *E. coli* could slowly utilize (doubling time of 11.2 ± 0.6 h; over the course of 72 h of fermentation in M9 malonate medium) malonate as a carbon source only at weak expression level (Fig. 4C). In contrast to complete utilization of malonate by *P. aeruginosa* within 12 h of fermentation (Fig. 4C), the observed slow utilization by *E. coli* suggests a potential metabolic imbalance in *E. coli*, perturbed by and unable to accommodate the malonate pathway.

Therefore, we implemented adaptive laboratory evolution to allow *E. coli* to rebalance and optimize its metabolism with malonate as a substrate. The *E. coli* strain carrying the pMdcR_iM was grown in an M9 malonate medium and evolved using serial passaging that imposes growth rate selection pressure (Fig. 4D) on an automated ALEbot[48]. After 21 passages, populations showed faster growth with a short lag phase compared to their ancestor (Fig. 4E). The evolved populations fully consumed malonate within 40 hrs. Subsequently, three clones were isolated from each replicate evolved population, and they all displayed a faster growth rate than the ancestor (Fig. 4F).

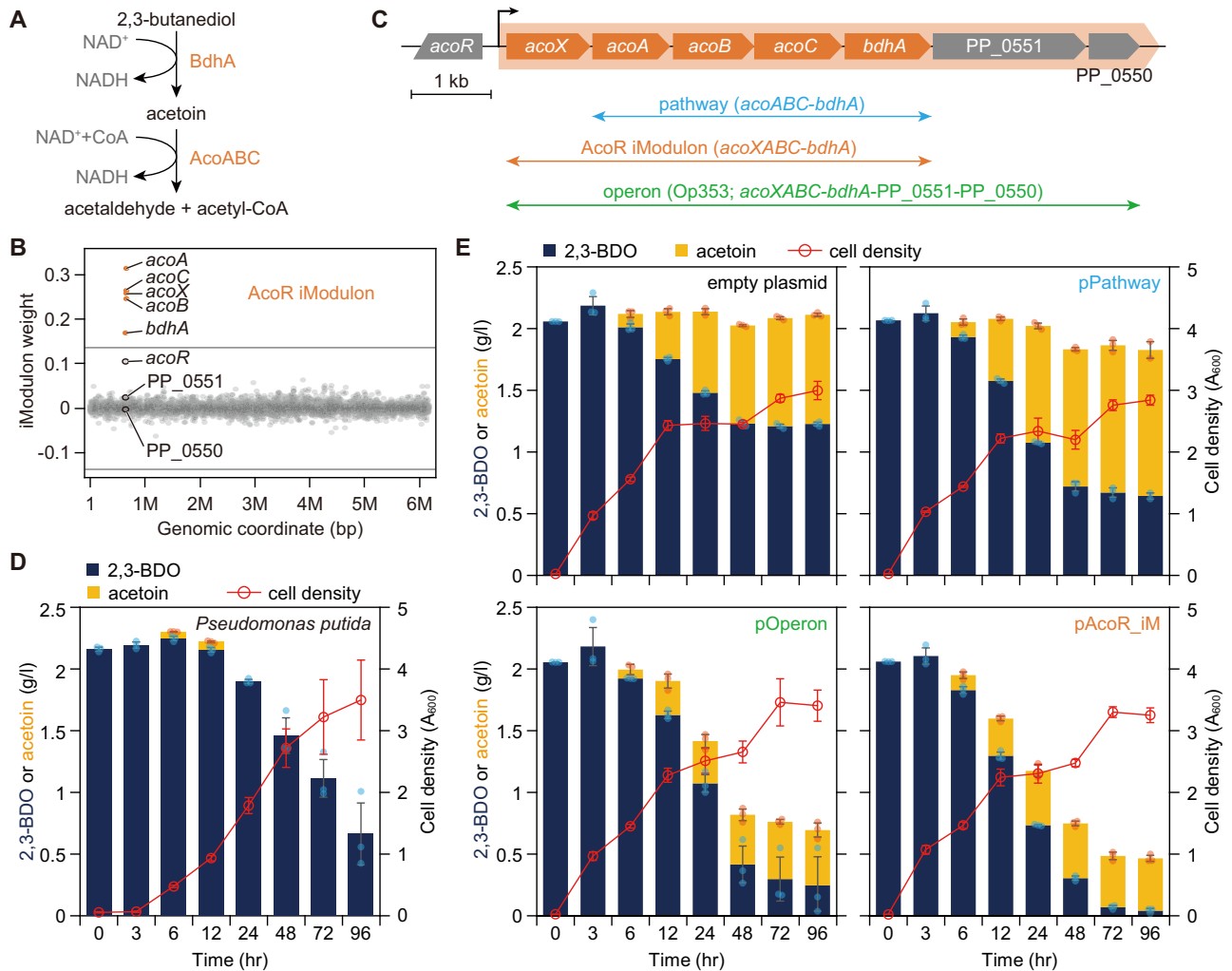

**Fig. 3 | Cross-species transfer of 2,3-butanediol (2,3-BDO) utilization iModulon of *Pseudomonas putida* in *E. coli*. A** A pathway responsible for 2,3-BDO utilization. **B** Scatter plot shows weights of genes in *P. putida* to AcoR iModulon. Gray lines indicate thresholds for determining iModulon membership. Five genes constitute the AcoR iModulon (orange circles). Gray circles identify genes not in the iModulon. Black circles are three neighboring genes. **C** Genomic structure of the AcoR iModulon. Orange shade shows predicted operonic structure. Genes in the iModulon are in orange. Arrows indicate three different plasmid constructs for cross-species transfer. **D** 2,3-BDO degradation by *P. putida*. The formation of acetoin was negligible. Blue and yellow boxes represent 2,3-BDO and acetoin in the culture medium. Red circles show cell density. Dots indicate individual data points. Data were presented as mean values ± SD. Error bars indicate the SD of the three biological replicates. **E** 2,3-BDO and acetoin degradation by *E. coli* carrying empty plasmid or one of the three constructs. 2,3-BDO was added at the start of the culture and the remaining amount and acetoin formation was measured. Blue and yellow boxes represent 2,3-BDO and acetoin in the culture medium. Red circles show cell density. Dots indicate individual data points. Data were presented as mean values ± SD. Error bars indicate SD of the three biological replicates. Source data are provided as a Source Data file.

To understand the genetic bases of improved growth, we resequenced the genome of the evolved clones (Supplementary Table 1). All the evolved clones carried mutations on DNA polymerase I, encoded by *polA*, which is required for plasmid maintenance (Fig. 4G)[49]. Previous studies reported a change of plasmid copy number induced by *polA* mutation[50]. Quantitative measurement of plasmid copy number indicated a reduction of plasmid copy number, which led to a reduction of MdcR iModulon expression (Fig. 4H). Thus, the initial metabolic failure was likely due to the sub-optimal expression of the MdcR iModulon (Supplementary Note, section 2), which could be optimized by ALE.

Engraftment of the MdcR iModulon, in addition to three other iModulons, demonstrated cross-species iModulon transfer as a rapid way of creating new functionality in bacteria with minimal engineering. We found that the overall behavior of the iModulon interferes with the host factors that require modifications to optimally support the system. This optimization could be rapidly achieved by ALE that identified

few genetic changes in the host, while the transferred genes acquired no adaptive mutations.

## Discussion

Historically synthetic biology has built multigenic functions in a serial manner. Individual genes are introduced to the host, and their function is assessed. iModulons, obtained through big data analysis of transcriptome compendia, describe sets of co-expressed genes that constitute independent cellular functions, suggesting that multigenic traits can be captured and transferred. Here we demonstrate that this is possible through cross-species transfer of cellular functions from *Pseudomonas* species into *E. coli*. We successfully transfer three metabolic traits and an antimicrobial resistance trait between these Gram-negative species.

iModulons lead to the identification of genes that are not functionally annotated but essential for recreating a cellular function in a new host. Thus, these genes gain a network-level functional

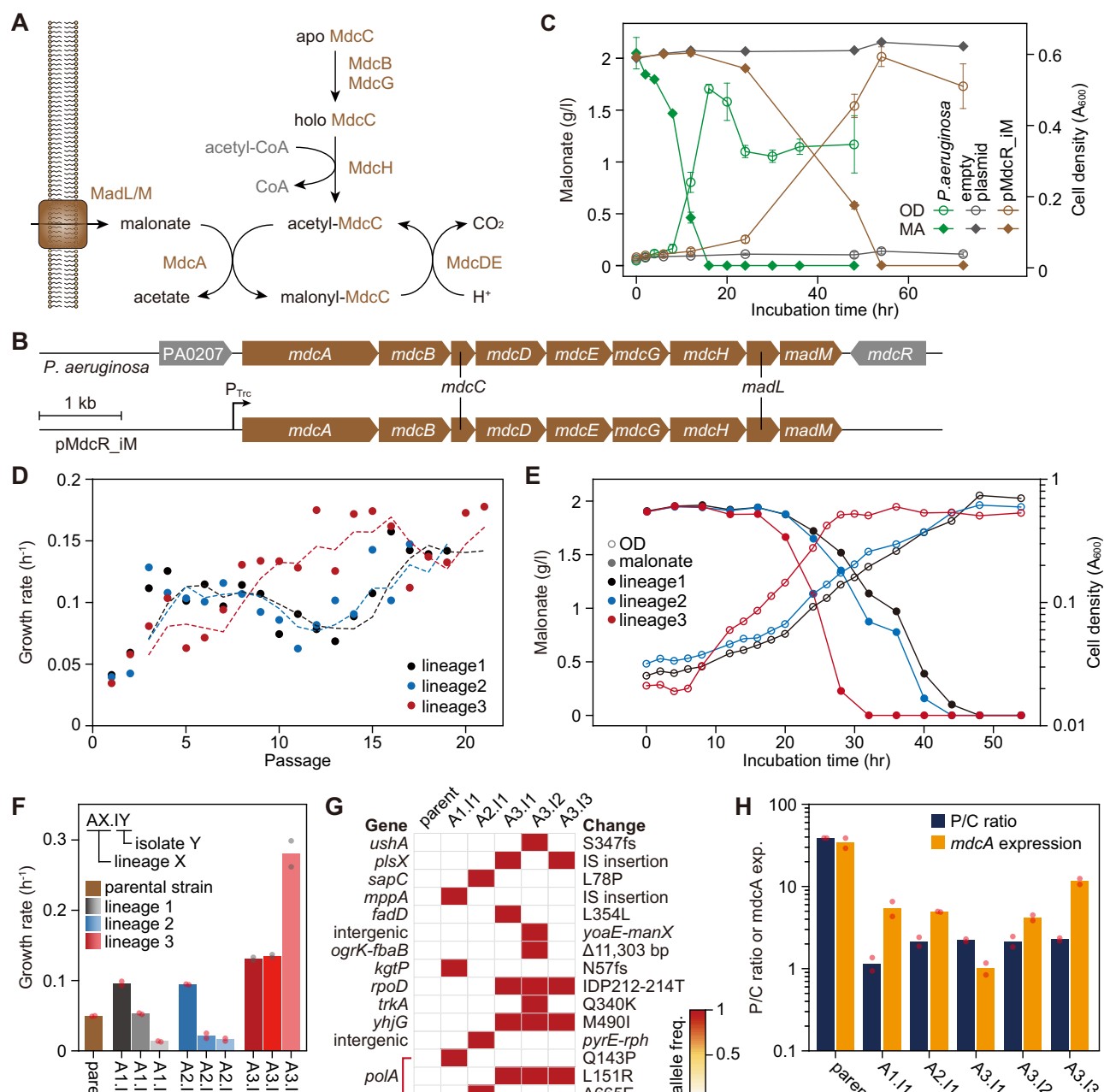

**Fig. 4 | Adaptive laboratory evolution improves functionality of *Pseudomonas aeruginosa* MdcR iModulon in *E. coli*. A** Malonate catabolic pathway in *P. aeruginosa*. **B** Genetic structure of the malonate catabolic operon of *P. aeruginosa* cloned in a heterologous expression plasmid, pMdcR_iM. Brown genes constitute MdcR iModulon. **C** Malonate utilization of *P. aeruginosa*, *E. coli* carrying empty plasmid, and pMdcR iM. Cells were incubated for up to 72 h in M9 malonate (2 g/l) media. Circles and diamonds show cell density and malonate concentration in culture, respectively. Green, gray, and brown lines represent *P. aeruginosa*, *E. coli* carrying empty plasmid, and pMdcR iM plasmid, respectively. Data were presented as mean values ± SD. Error bars indicate the SD of three replicated cultures. **D** Growth rates of *E. coli* carrying the MdcR iModulon over the course of evolution. Dashed lines are moving averages of three individual ALE lineages. Growth rates for each ALE lineage are colored differently. **E** Malonate utilization and growth of three evolved populations. Circles represent cell density, with the solid circles being

extracellular malonate concentrations. Measurements for each ALE lineage are colored differently. Data were presented as mean values of two replicated cultures. **F** Growth rates of clones isolated from malonate-evolved populations in M9 malonate medium. Data were presented as mean values of two replicated cultures. Dots show individual data points. Strain names are given as AX.IY. X is the ALE lineage number and Y is an arbitrary identifying number for the clonal isolate from the same ALE lineage. **G** Adaptive mutations in the ALE endpoint clones that are not present in the parental strain. fs, frameshift mutation. The heatmap shows allele frequencies colored as in the provided color key. **H** Plasmid-to-chromosome copy number ratio (P/C ratio) and expression level of *mdcA* of unevolved parent strain and evolved clones. Blue and orange boxes represent the P/C ratio and *mdcA* expression level, respectively. Data were presented as mean values in two biologically replicated cultures. Dots are individual data points, each of which is composed of two technical duplicates. Source data are provided as a Source Data file.

annotation, as opposed to classical molecular-level functional annotation. We also find that iModulon genes may have an auxiliary function, i.e., they enhance the cellular functions being transferred. Finally, we find that laboratory evolution can improve the transferred cellular function in the new host. Sequencing laboratory-evolved strains can identify host factors that enable or enhance the cellular function encoded on the transferred construct.

Taken together, these factors lead to an advancement of synthetic biology, including the identification of all genes needed to constitute a cellular function and revealing host factors that need modification to optimize the engineered strain. This acceleration and higher predictability in strain design and construction should accelerate the development of synthetic biology and its deployment for practical purposes such as biomanufacturing.

## Methods

### Bacterial strains and culture conditions

*E. coli* strain K-12 substrain MG1655 is used as a recipient of iModulon. Cells were grown in LB medium (Novagen, 71753) or M9-defined medium (47.75 mM $Na_2HPO_4$, 22.04 mM $KH_2PO_4$, 8.56 mM NaCl, 18.70 mM $NH_4Cl$, 2 mM $MgSO_4$, 0.1 mM $CaCl_2$, and trace elements). Trace elements were prepared in 2000× concentrated solution (100 mM $FeCl_3$, 9.54 mM $ZnCl_2$, 8.41 mM $CoCl_2$, 8.27 mM $Na_2MoO_4$, 0.75 mM $CaCl_2$, 0.91 mM $CuCl_2$, and 0.5 mM $H_3BO_3$ in 3.7% (w/w) hydrochloric acid solution). Branched-chain amino acids were supplemented with a final combined concentration of 0.15 mM. To construct the ΔBCAA strain (MG1655 Δ*ilvLGMEDAYC*::*ilvY*, Δ*leuLABCD*, *thrL-119*::*sacB-cat*), sequential lambda-recombinations were performed[51]. First, *ilvLGMEDAYC* was replaced with a kanamycin resistance cassette flanked by two flippase recognition sequences. The cassette was removed by flippase-mediated excision, and a DNA cassette containing *ilvY-pheS*\*-npt*II was introduced to replace the flippase recombination scar. Then *pheS*\*-npt*II dual-selectable marker was again removed by the lambda recombination. A successful recombinant was selected by *p*-chloro-phenylalanine counter selection[52] on an LB-agar plate containing 2 mM *p*-chloro-phenylalanine. The genomic locus containing *leuLABCD* was removed by introducing the kanamycin cassette, followed by flippase recombination. Lastly, a *sacB-cat* dual selection marker was inserted upstream of *thrL* serving as a landing pad for REXER. Yeast was grown in YPDA (10 g/l yeast extract, 20 g/l peptone, 30 mg/l adenine sulfate, and 20 g/l dextrose) or synthetic His dropout medium (1.7 g/l yeast nitrogen base without amino acid and ammonium sulfate, 1.9 g/l yeast synthetic drop-out medium supplements without histidine, 5 g/l ammonium sulfate, 20 g/l agar, and 20 g/l dextrose) at 30 °C. MdcR and AmpC iModulons were sourced from *Pseudomonas aeruginosa* strain K2733[16]. AcoR and VanR iModulons were amplified from *Pseudomonas putida* strain KT2440. The fermentation experiment was done in a 15 ml liquid culture contained in a 30-ml test tube. Cells were incubated at 37 °C on a heatblock and stirred at 600 rpm with a cross-shaped magnetic stir bar. To induce heterologous iModulons, (IPTG) was added to cultures to 0.1 or 1 mM of final concentration. All reagents were purchased from Sigma-Aldrich unless stated otherwise.

### Investigation of target iModulons and transfer design

Target iModulons for transfer were selected from a catalog of iModulons available through iModulonDB (https://imodulondb.org/)[12]. Initially, iModulons of *P. aeruginosa* and *P. putida* were inspected by their predicted functional annotations. iModulons associated with functions that are uncharacterized, already present in *E. coli* (e.g., translation), or challenging to experimentally screen (e.g., quorum sensing) were excluded. Subsequently, the functions of iModulons were further investigated by a comprehensive examination of individual gene functions using gene annotation databases (Biocyc[18] and Pseudomonas Genome Database[53]) and a literature search.

The functional annotation of an iModulon was revised in this step if needed. For instance, the *P. aeruginosa* MdcR iModulon was initially annotated as a malonate biosynthetic process, but we corrected its function to malonate catabolism as it encodes the malonate catabolic enzyme complex[21]. Finally, iModulon activities across diverse experimental conditions were examined using the built-in graph function of iModulonDB in the individual iModulon page, which can also be obtained by downloading two tables, namely Experimental Conditions and iModulon Activity. In this step, we took a rational approach to test if changes in iModulon activity under specific conditions were consistent with established biological knowledge. For example, activities of BCAA biosynthetic iModulons and AmpC iModulon are induced in the absence of BCAA and the presence of beta-lactam in the media (Supplementary Figs. 1C, 2), affirming the functional annotations.

The strategy for iModulon transfer was established by addressing the genetic context of iModulons. Firstly, gene members of iModulons were extracted by selecting genes with their absolute iModulon weight exceeding the threshold (Figs. 1B, 2A, 3B and Supplementary Figs. 1A, 4) (Gene Table in an iModulon page). Next, the genetic organization (location, orientation, and operonic structure) was assessed through Biocyc[18] and Pseudomonas Genome Database[53]. If genes were found in a single operon, the entire operon was directly amplified and cloned. If genes were located in multiple genomic loci, they were refactored into an operon, while preserving the genetic order to ensure an optimal expression level, as demonstrated previously[25,26].

### DNA assembly and cloning for capturing iModulon

The bacterial artificial chromosome (BAC) encoding BCAA biosynthetic iModulon was constructed by assembling PCR amplified pCAP_BAC (Addgene plasmid #120229)[54] fragments, *npt*II promoter, *pheS-npt*II, *thr*, *ilvG*, *ilvMEDA*, *ilvC*, and *leu* loci using transformation-associated recombination (TAR) cloning[55] in *Saccharomyces cerevisiae* strain VL6-48 (ATCC MYA-3666; MATα his3-Δ200 trp1-Δ1 ura3 − 52 lys2 ade2 − 1 met14 cir0). To perform TAR cloning, yeast VL6-48 was grown in 50 ml YPDA broth at 30 °C until $A_{600}$ of 0.4. Cells were collected by centrifuging at 3000 × g for 5 min. Then the cell pellet was washed with 10 ml of buffer 1 (100 mM LiAc, 10 mM Tris-HCl (pH8.0), and 1 mM EDTA (pH8.0)) and resuspended with 0.5 ml of buffer 1. Cell suspension (100 μl) was combined with 100 ng each of the DNA fragments, 100 μg of denatured salmon sperm DNA (Sigma-Aldrich, D9156), and 600 μl of buffer 2 (100 mM LiAc, 10 mM Tris-HCl (pH8.0), 1 mM EDTA (pH8.0), 40% (w/v) PEG-3350 (Sigma-Aldrich, P4338)). The mixture was mixed thoroughly and incubated at 30 °C for 30 min. Then, the mixture was incubated at 42 °C for 15 min after the addition of 70 μl of DMSO. The cell was harvested by centrifugation at 16,000 × g for 1 min after incubation on ice for 1 min. Liquid supernatant was discarded, and the cell pellet was resuspended with 1 ml of YPDA broth. After incubation at 30 °C for 2 h, the cell was harvested again with centrifugation at 16000 × g for 1 min and resuspended with 100 μl of sterile water. Successful clones were screened on a solid synthetic His dropout medium. A spacer array targeting both the landing pad and the BAC was constructed by annealing of two primers, RX4_1_F and RX_4_1_R followed by primer extension using RX4_2_F and RX4_2_R. The spacer array was cloned in a pUC plasmid containing the crRNA leader sequence. MdcR and AcoR iModulons were PCR amplified from *P. aeruginosa* and *P. putida* genomic DNA. DNA fragments had 15−20 nt homology to pTrcHis2A (Invitrogen, V36520) plasmid and cloned into the linearized plasmid backbone using In-Fusion Cloning Kit (Takara Bio, 638948). Two converging operons encoding VanR iModulons were PCR amplified separately from *P. putida* genomic DNA and cloned into pTrcHis2A plasmid backbone in three-fragments ligation reaction using In-Fusion Cloning Kit resulting in pVanR_iM. AmpC iModulon was PCR amplified as six different fragments from *P. aeruginosa* genomic DNA. Together with the Trc promoter fragment amplified from pTrcHis2A plasmid and pCAP_BAC, entire DNA fragments were

assembled using TAR cloning. Assembled BAC was extracted using Gentra Puregene Yeast/Bact. Kit (Qiagen, 158567). Plasmids and BACs were electro-transformed to *E. coli* NEB10β (New England Biolabs, C3020K) for propagation and MG1655 for downstream experiments. Primer sequences for PCR amplification were summarized in Supplementary Table 2. All the plasmids were sequence confirmed by whole-plasmid sequencing (Primordium Labs).

### REXER

The refactored BCAA biosynthetic iModulon was introduced into the genome using the replicon excision enhanced recombination (REXER) method[56,57] with minor modification. First, BAC_BCAA_Biosynthetic_iM plasmid containing BCAA biosynthetic iModulons was introduced using electroporation to *E. coli* ΔBCAA strain carrying pREDCas9 plasmid and *sacB-cat* dual-selection landing pad upstream of *thrL* gene. The recipient cell was grown in terrific broth (TB) containing 25 μg/ml chloramphenicol, 50 μg/ml kanamycin, 100 μg/ml carbenicillin, and 30 mM L-arabinose at 37 °C to cell density ($A_{600}$) of 0.8. Two micrograms of PCR-amplified linear guide-RNA construct, targeting both the landing pad and the BAC, were introduced using electroporation, and cells were screened on NaCl-free LB-agar plate (10 g/l tryptone, 5 g/l yeast extract, and 1.5% agar) containing 100 μg/ml carbenicillin, 50 μg/ml kanamycin, and 10% sucrose. Successful recombinants were further screened by chloramphenicol sensitivity and PCR genotyping.

### HPLC

Culture supernatant was collected by filtering 200 μl of the crude culture through a 96-Well PVDF Filtration Plate (0.2 μm; Agilent, 203980-100). Exo-metabolites were analyzed from 10 μl of the sample using 1260 Infinity II HPLC System (Agilent) equipped with Multi-sampler (G7167A), HIP Degasser (G4225A), Binary Pump (G1312C), and Refractive Index Detector (G1362A), Thermostatted Column Compartment (G1316A), and Aminex HPX-87H HPLC Column (300 × 7.8 mm; BioRad, 1250140). Five millimolar sulfuric acid was used as a mobile phase, and the detector temperature was maintained at 30 °C. Vanillate, protocatechuate, acetoin, and 2,3-butanediol were detected at a column temperature of 65 °C with a mobile phase flow rate of 0.6 ml/min. For detection of malonate column temperature was maintained at 45 °C.

### Quantitative PCR

To measure the plasmid-to-chromosome ratio and the expression level of the mdcR iModulon construct, cells were grown in 15 ml M9 malonate medium and sampled at the mid-log phase. Total DNA was extracted from 1 ml of the culture using a Quick-DNA Miniprep Kit (Zymo Research, D3024) as instructed by the manufacturer. 100 ng of DNA extract was subject to quantitative PCR in a 20 μl reaction containing AccuPower PCR 2× Master Mix (Bioneer, K-2018), 10 μM each of primers, SYBR Green I Nucleic Acid Gel Stain (Invitrogen, S7563). Fluorescence signals were monitored by the CFX Duet Real-Time PCR System (RioRad, 12016265). The amount of beta-lactamase (*bla*) gene on the plasmid was compared to the *alaA* gene on the chromosome to estimate the relative number of the plasmid-to-chromosome. PCR efficiencies were measured from three twofold dilution series of each gene. Plasmid-to-chromosome ratio (P/C ratio) is calculated by

$$P/C\ ratio = \frac{E_{alaA}^{Cq \cdot alaA}}{E_{bla}^{Cq \cdot bla}} \qquad (1)$$

where $E_{alaA}$ and $E_{bla}$ are the PCR efficiencies of *alaA* gene and *bla* gene, respectively. $Cq \cdot alaA$ and $Cq \cdot bla$ are the quantification of cycles of respective genes.

To estimate the expression level of MdcR iModulon, the relative expression level of *mdcA* to 16S rRNA was measured. RNA was extracted from 14 ml of the culture using Quick-RNA Fungal/Bacterial

Miniprep Kit (Zymo Research, R2014) as instructed by the manufacturer. Residual DNA was removed by incubating 2 μg of RNA extract at 37 °C for 30 min in a 50 μl reaction containing 2 U of RNase-free DNase I (New England Biolabs, M0303) followed by a purification using RNA Clean & Concentrator Kit (Zymo Research, R1013). cDNA was synthesized from 300 ng of the DNA-depleted RNA sample using the SuperScript II First-Strand Synthesis System (Invitrogen, 11904018) as instructed by the manufacturer. About 1 μl of cDNA synthesis reaction was subject to quantitative PCR in a 20 μl reaction containing Accu-Power PCR 2× Master Mix, 10 μM each of primers, SYBR Green I Nucleic Acid Gel Stain. The relative amount of *mdcA* transcript was quantified by the ΔΔCq method using *rrsA* transcript as a reference, after adjusting PCR efficiencies that were measured from three twofold dilution series of each gene. All the primers were designed by Primer-BLAST[58] to have no predicted cross-reactivity. Primer sequences are summarized in Supplementary Table 2.

### Antibiotics sensitivity assay

For disc diffusion assay, overnight grown *E. coli* culture was diluted to $5 \times 10^8$ cells/ml and 500 μl of the diluted culture was spread on an LB-agar plate. Thirty microliters of ampicillin solutions with different concentrations were dropped on sterilized filter paper disks (9 mm diameter; Sigma-Aldrich, 1703932). After drying for 15 min, the disks were placed on the LB-agar plate and the plate was incubated at 37 °C overnight. For the dose-killing assay, overnight grown *E. coli* culture was diluted to $5 \times 10^8$ cells/ml in LB medium containing an appropriate concentration of ampicillin, and 100 μl of the diluted culture was transferred to a 96-well microplate. Cells were incubated at 37 °C with agitation in Infinite 200 Pro microplate reader (Tecan) and $A_{600}$ was monitored every 15 min up to 10 h.

### Adaptive laboratory evolution

*E. coli* K-12 MG1655 carrying pMdcR_iM plasmid was evolved via serial propagation of 150 μl into 15 ml M9 malonate (2 g/l) minimal medium containing 100 μg/ml carbenicillin in three biologically replicated cultures. Cultures were incubated at 37 °C, aerated by magnetic stirring, and reinoculated at $A_{600}$ of 0.3 (Tecan Sunrise microplate reader; equivalent to an $A_{600}$ of 0.750 on a conventional spectrophotometer with a path length of 10 mm) using an automated system. After 40 days of propagation, evolved cultures were subjected to a malonate utilization assay, and three clones were isolated from each culture.

### DNA sequencing and analysis

Genomic DNA was isolated using Quick-DNA Miniprep Kit as instructed by the manufacturer. Whole-genome DNA-seq libraries were generated with a NEBNext Ultra II DNA Library Prep Kit for Illumina (New England Biolabs, E7645) and run on an Illumina NovaSeq X Plus with 100 cycles pair-ended recipe. The sequencing results were processed with the in-house pipeline[59] that incorporates the BreSeq pipeline to identify mutations[60]. *E. coli* K-12 MG1655 genome sequence (National Center for Biotechnology Information accession no. NC_000913.3) was used as a reference sequence.

### Reporting summary

Further information on research design is available in the Nature Portfolio Reporting Summary linked to this article.

## Data availability

The genome resequencing data generated in this study have been deposited in the EMBL Nucleotide Sequence Database (ENA) under accession code PRJEB69310. The iModulon activity and gene membership data used in this study are available in the iModulonDB [https://imodulondb.org/]. Predicted operonic structures were from the Biocyc database [https://www.biocyc.org/]. Genome sequence and genetic organization information were assessed through the

 

Pseudomonas Genome Database [https://www.pseudomonas.com/]. All other data are available in the article, Supplementary items, or Source Data file. Source data are provided with this paper.

## Code availability

The mutation detection from the DNA sequencing results was performed with the ALEdb (v1.1.0) pipeline [https://www.aledb.org/] that incorporates the BreSeq pipeline[59,60].

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

## Acknowledgements

This work was funded by the Novo Nordisk Foundation grant NNF20CC0035580 and the Y.C. Fung Endowed Chair in Bioengineering at UC San Diego to B.O.P. We thank Ms. Emily Armbruster and Dr. Joe Pogliano for providing *Pseudomonas aeruginosa* strain K2733. We thank Marc Abrams for editing the manuscript.

## Author contributions

B.O.P. conceived and supervised the study. D.C., A.M.F., and B.O.P. designed the experiments. D.C., C.A.O., R.S., H.Y. and J.S. performed the experiments. D.C., A.M.F. and B.O.P. analyzed the data. D.C., A.M.F. and B.O.P. wrote the manuscript. All authors read and approved the final manuscript.

## Competing interests

The authors declare no competing interests.
