## [Peer Review File · Nature Communications]

Reviewers' Comments:

Reviewer #1:

Remarks to the Author:

The manuscript from Choe & colleagues tackles the issue of transferring cellular function between bacterial species. They do so by investigating the added value of using iModulons (as opposed to genome annotation or operon selection) to predict genes that are necessary and sufficient to carry a given cellular function. The authors demonstrate this added value via several use cases of transferring metabolic functions and antimicrobial resistance from *Pseudomonas putida* and *aeruginosa* to *E. coli*. The authors also use serial propagation to successfully improve the phenotype of a strain after function transfer.

While the paper is a follow up of previous publications on iModulon by the some of the same authors (refs. 10, 12, 14-16 and others), the reported work does demonstrate the added value of iModulons for the prediction of genes that are relevant to cellular functions.

I have three main comments about: (1) the general structure of the article; (2) the focus of the claims; and (3) a potential overlook. These should be taken account prior publication.

(1) Concerning the general structure:

Within the Results section, the train of thoughts (in terms of the questions successively addressed by the authors) can be confusing. What the authors do, roughly speaking:

- (i) They show that iModulon transfer can work (via the VanR example).
- (ii) They show that, subsequently to iModulon transfer, they can use laboratory evolution to improve cellular function and highlight investigation routes (via the MdcR example).
- (iii) They show that the transfer of either more or less genes (by way of operon or gene annotation, respectively) can be outperformed by the transfer of an iModulon (via the AcoR example).
- (iv) They show that the transfer of less genes can also be outperformed by the transfer of an iModulon, even for non-metabolic functions (via the AmpC example).

The train of thoughts would be clearer and more appealing if the authors successively addressed the questions related to the added values of iModulons. Here is a possible layout example:

- (i) Can the transfer of an iModulon outperform seemingly simple cases of non-metabolic functions such as antimicrobial resistance? (via AmpC example)
- (ii) Can the transfer of an iModulon outperform the alternative methods of operon selection or gene annotation for metabolic functions? (via AcoR example)
- (iii) Can the transfer of an iModulon be improved by laboratory evolution? (via MdcR example).

As presented in the manuscript, the VanR example is not answering a specific question the authors may phrase a specific question of what they wanted to achieve with VanR or relegate the example to supplementary materials.

Readjusting some of the article's architecture would alleviate the impression sometimes left by the article of cataloguing work that was done, instead of addressing scientific questions.

(2) Concerning the general claims of the article:

The authors emphasize that iModulon "enables cross-species transfer of cellular functions with a single transformation". This is misleading, as this is enabled by molecular biology techniques, regardless of how the genes were selected.

A more appropriate claim is that iModulons allow to better identify which genes are necessary and sufficient to transfer a cellular function.

(3) Concerning the phenotypic improvement via laboratory evolution:

The authors attempt laboratory evolution to improve the phenotype of the MdcR iModulon. A simpler and faster and potentially more effective approach could have been the following: instead of cloning all the iModulon genes in a new operon, the authors could have (at least partially) preserved the native genetic arrangement (i.e. the relative orientation of genes). There is indeed ample evidence in the literature that the relative orientation of genes plays a crucial role in determining their relative gene expression levels (Nagy-Staron et al. 2021 eLife; Boulas et al. 2023 Nucleic Acids Res.).

In fact, the authors even conclude via that their laboratory evolution yielded mutations of polymerase I and potentially plasmid copy number, suggesting that gene expression levels were the issue.

It could thus be helpful that the authors mention how a preservation of genetic architecture might improve iModulon transfer.

Other minor comments

- Line 92: Four genes, vanAB, galP, and vanK should be vanA, vanB, galP-IV, and vanK
- Line 109: 109 ALE optimizes the functionality of catabolic - iModulons does not appear as a subsection but should.
- It would be nice to find in the Methods section some explanation on iModulon usage for the four provided examples. To that end, some information found in supplementary materials could be moved to Methods

Reviewer #2:

Remarks to the Author:

This article reports a transcriptomics database tool platform iModulonDB for the holistic heterologous expression of cross-species cell functions. The iModulonDB platform effectively guides expression of various biological functions in different species. The Palsson team demonstrates the extensive applications of this platform in strain design and synthetic biology research through four case studies, including the heterologous expression of *Pseudomonas putida*'s vanillin-to-protocatechuic acid conversion function in *Escherichia coli*, *Pseudomonas aeruginosa*'s malonic acid metabolism function, *Pseudomonas putida*'s 2,3-butanediol metabolism function, and *Pseudomonas aeruginosa*'s ampicillin resistance.

The iModulonDB platform contains a wealth of transcriptomics information, allowing researchers to easily identify genes that may be associated with specific functions of interest in the original host. These genes are overexpressed at the transcriptomics level during the execution of these functions and have been found to be correlated. Some of these genes may have been previously studied in relation to the target function, while others have not received any precedent notice. The work is suitable for the journal.

Suggestions:

1. It is recommended that the control data of the original strain be included in the figures showing cross-species functional restoration experiments. This will allow a more rigorous and intuitive demonstration of the effectiveness of heterologous strains in replicating functions. If relevant data are already available, please directly cite them; otherwise, it is suggested to conduct supplementary experiments.
2. The description of laboratory adaptive evolution (ALE) in this article has flow inconsistency. Moving it to the latter part of the article may better connect with the discussion section.

Point by point response to reviewers

Reviewer #1

Comments for the Authors

The manuscript from Choe & colleagues tackles the issue of transferring cellular function between bacterial species. They do so by investigating the added value of using iModulons (as opposed to genome annotation or operon selection) to predict genes that are necessary and sufficient to carry a given cellular function. The authors demonstrate this added value via several use cases of transferring metabolic functions and antimicrobial resistance from *Pseudomonas putida* and *aeruginosa* to *E. coli*. The authors also use serial propagation to successfully improve the phenotype of a strain after function transfer.

While the paper is a follow up of previous publications on iModulon by the some of the same authors (refs. 10, 12, 14-16 and others), the reported work does demonstrate the added value of iModulons for the prediction of genes that are relevant to cellular functions.

I have three main comments about: (1) the general structure of the article; (2) the focus of the claims; and (3) a potential overlook. These should be taken account prior publication:

Main comments:

1. Concerning the general structure: Within the Results section, the train of thoughts (in terms of the questions successively addressed by the authors) can be confusing. What the authors do, roughly speaking:

- (i) They show that iModulon transfer can work (via the VanR example).
- (ii) They show that, subsequently to iModulon transfer, they can use laboratory evolution to improve cellular function and highlight investigation routes (via the MdcR example).
- (iii) They show that the transfer of either more or less genes (by way of operon or gene annotation, respectively) can be outperformed by the transfer of an iModulon (via the AcoR example).
- (iv) They show that the transfer of less genes can also be outperformed by the transfer of an iModulon, even for non-metabolic functions (via the AmpC example).

The train of thoughts would be clearer and more appealing if the authors successively addressed the questions related to the added values of iModulons. Here is a possible layout example:

- (i) Can the transfer of an iModulon outperform seemingly simple cases of non-metabolic functions such as antimicrobial resistance? (via AmpC example)
- (ii) Can the transfer of an iModulon outperform the alternative methods of operon selection or gene annotation for metabolic functions? (via AcoR example)
- (iii) Can the transfer of an iModulon be improved by laboratory evolution? (via MdcR example).

As presented in the manuscript, the VanR example is not answering a specific question the authors may phrase a specific question of what they wanted to achieve with VanR or relegate the example to supplementary materials.

Readjusting some of the article's architecture would alleviate the impression sometimes left by the article of cataloging work that was done, instead of addressing scientific questions.

Response: We appreciate the valuable suggestion from the reviewer and have incorporated the recommended reorganization to enhance the logical flow of our manuscript. Following the reviewer's advice, we have reordered the presentation of iModulon transfer examples to (i) VanR, (ii) AmpC, (iii) AcoR, and (iv) MdcR. We hope this new sequence provides a more cohesive narrative and improves the overall readability of the manuscript. Please refer to the revised manuscript for a detailed exploration of these changes, including updates to section headings

RESULTS

1. Cross-species transfer of *Pseudomonas* iModulons into *E. coli*
2. Auxiliary genes may be needed for optimal function of cross-species transferred iModulons
3. Complete iModulon gene membership is needed for successful cross-species transfer
4. ALE optimizes the functionality of catabolic iModulons

We have opted to maintain the VanR iModulon example in the main text to serve as a straightforward illustration of iModulon's capabilities — the rapid identification of biological processes. This inclusion aims to familiarize the audience with the concept of iModulon transfer. However, we acknowledge the reviewer's concern and have revised the VanR section in the manuscript to underscore the crucial point that iModulons enable the swift identification and transfer of biological processes.

P3, L85-87: First, we chose to reconstruct and transfer a simple bioconversion process from *Pseudomonas putida*¹⁵ to *E. coli* in order to examine iModulon's capability to rapidly identify genes associated with specific function.

P4, L105-111: *E. coli* carrying pVanR_iM converted VA into PCA up to 15.34 mg/l passively diffused to the supernatant^{27,28} during 48 hours of fermentation in M9 glucose (4 g/l) medium supplemented with 100 mg/l VA, while the negative control carrying empty plasmid did not metabolize any VA (**Fig. 1E**). This first cross-species iModulon transplantation illustrates the rapid identification of enzymes required for biotransformation by ICA. Furthermore, iModulon engraftment provided a rapid way to biochemically verify a predicted pathway in a heterologous host.

2. Concerning the general claims of the article: The authors emphasize that iModulon "enables cross-species transfer of cellular functions with a single transformation". This is misleading, as this is enabled by molecular biology techniques, regardless of how the genes were selected.

A more appropriate claim is that iModulons allow to better identify which genes are necessary and sufficient to transfer a cellular function.

Response: We agree with the reviewer's comment that the primary objective of this manuscript is to demonstrate the capability of iModulons in accurately identifying genes necessary and sufficient to constitute cellular functions, irrespective of specific hosts. Accordingly, we have made revisions to the manuscript as outlined below:

P1, L21-23: The identification of iModulons enables accurate identification of genes necessary and sufficient for cross-species transfer of cellular functions ~~with a single transformation~~.

P2, L49-50: This advancement opens up the possibility to identify targeted functions in a particular strain and transfer them into alternate hosts ~~single transformation~~.

P9, L286-289: iModulons, obtained through big data analysis of transcriptome compendia, describe sets of co-expressed genes that constitute independent cellular functions, suggesting that multigenic traits can be captured and transferred ~~in a single transformation~~.

P9, L302-304: Taken together, these factors lead to an advancement of synthetic biology, including identification of all genes needed to constitute a cellular function, ~~enablement of a single transformation event~~, and revealing host factors that need modification to optimize the engineered strain.

3. Concerning the phenotypic improvement via laboratory evolution:

The authors attempt laboratory evolution to improve the phenotype of the MdcR iModulon. A simpler and faster and potentially more effective approach could have been the following: instead of cloning all

the iModulon genes in a new operon, the authors could have (at least partially) preserved the native genetic arrangement (i.e. the relative orientation of genes). There is indeed ample evidence in the literature that the relative orientation of genes plays a crucial role in determining their relative gene expression levels (Nagy-Staron et al. 2021 eLife; Boulas et al. 2023 Nucleic Acids Res.).

In fact, the authors even conclude via that their laboratory evolution yielded mutations of polymerase I and potentially plasmid copy number, suggesting that gene expression levels were the issue.

It could thus be helpful that the authors mention how a preservation of genetic architecture might improve iModulon transfer.

Response: As highlighted by the reviewer, the order and orientation of genes can have a profound impact on gene expression, influenced by various molecular mechanisms such as transcriptional read-through and DNA topology, as extensively studied in previous reports (Nagy-Staron, A. et al. *Local genetic context shapes the function of a gene regulatory network. Elife 10, (2021).* Boulas, I. et al. *Assessing in vivo the impact of gene context on transcription through DNA supercoiling. Nucleic Acids Res. 51, 9509–9521 (2023)*). We fully acknowledge the significance of maintaining the genetic arrangement of iModulons, and we made every effort to preserve this arrangement whenever possible. It appears there might be a misunderstanding regarding one of our examples. In the mentioned case, MdcR, and AcoR iModulon, we copied an operon without altering its genetic arrangement as they natively appear as operons, and we want to clarify that no change was made to the original genetic structure during the transfer process. In order to provide a clearer representation, we have revised the figure to elucidate the genetic organization of the MdcR iModulon in *P. aeruginosa* (Fig. 4B).

Figure 4. Adaptive laboratory evolution improves functionality of *Pseudomonas aeruginosa* MdcR iModulon in *E. coli*. (B) Genetic structure of the malonate catabolic operon of *P. aeruginosa* cloned in a heterologous expression plasmid, pMdcR_iM.

In the instances of VanR and AmpC iModulons, we made efforts to maintain the original gene order. However, in situations where genes were sourced from multiple loci, we needed to reorient some genes to ensure they could be expressed under a single promoter. The design criteria guiding our decisions in this regard have been elaborated in the revised manuscript.

P3, L101-103: When refactoring iModulons for heterologous expression, we tried to preserve native genetic arrangement, for VanR and following iModulons if possible, to ensure optimal expression levels of the gene members as demonstrated elsewhere^{25,26}.

P11, L359-364: Next, the genetic organization (location, orientation, and operonic structure) was assessed through Biocyc¹⁸ and Pseudomonas Genome Database⁵³. If genes were found in a single operon, the entire operon was directly amplified and cloned. If genes were located in multiple genomic loci, they were refactored into an operon, while preserving the genetic order to ensure an optimal expression level as demonstrated previously^{25,26}.

Minor comments:

4. Line 92: Four genes, *vanAB*, *galP*, and *vanK* should be *vanA*, *vanB*, *galP-IV*, and *vanK*

Response: We failed to correctly express gene nomenclature. We revised the manuscript accordingly as follows:

P3, L93-95: Four genes, *vanA*, *vanB*, *galP*, and *vanK* are functionally annotated to encode for vanillate O-demethylase oxidoreductase complex, outer-membrane porin, and a major facilitator superfamily transporter, respectively²².

5. Line 109: 109 ALE optimizes the functionality of catabolic - iModulons does not appear as a subsection but should.

Response: It is supposed to be a section heading, however we failed to do so while handling the manuscript. We have revised the issue and there are four subsections now in the revised manuscript.

RESULTS

1. **Cross-species transfer of *Pseudomonas* iModulons into *E. coli***
2. **Auxiliary genes may be needed for optimal function of cross-species transferred iModulons**
3. **Complete iModulon gene membership is needed for successful cross-species transfer**
4. **ALE optimizes the functionality of catabolic iModulons**

6. It would be nice to find in the Methods section some explanation on iModulon usage for the four provided examples. To that end, some information found in supplementary materials could be moved to Methods

Response: We appreciate the constructive feedback provided by the reviewer. We acknowledge that we failed to provide a sufficient explanation of iModulon usage for audiences unfamiliar with the iModulon dataset. All the expression data, gene weights, and list of genes consisting iModulon are readily available through iModulonDB (<https://imodulondb.org/>; *Rychel, K. et al. iModulonDB: a knowledgebase of microbial transcriptional regulation derived from machine learning. Nucleic Acids Res. 49, D112–D120 (2021)*). In the iModulon transfer process, target iModulons were selected from iModulonDB, initially screening for *P. aeruginosa* and *P. putida* iModulons, excluding those associated with uncharacterized or *E. coli*-present functions. The functions were further scrutinized by examining individual gene functions and correcting annotations if necessary. iModulon activities across diverse experimental conditions were assessed, ensuring consistency with established biological knowledge. The strategy for iModulon transfer involved extracting genes based on their iModulon weight, assessing genetic organization, and, when needed, refactoring genes into operons while preserving optimal expression order. We have included a detailed section in the method to clarify iModulon investigation, selection, and transfer. Please refer to the new section for an improved explanation of these processes as follows:

ONLINE METHODS

Investigation of target iModulons and transfer design

Target iModulons for transfer were selected from a catalog of iModulons available through iModulonDB (<https://imodulondb.org/>)¹². Initially, iModulons of *P. aeruginosa* and *P. putida* were inspected by their predicted functional annotations. iModulons associated with functions that are uncharacterized, already present in *E. coli* (e.g., translation), or challenging to experimentally screen (e.g., quorum sensing) were excluded. Subsequently, the functions of iModulons were further investigated by a comprehensive examination of individual gene functions using gene annotation databases (Biocyc¹⁸ and *Pseudomonas* Genome Database⁵³) and a literature search. The functional annotation of an iModulon was revised in this step if needed. For instance, the *P. aeruginosa* MdcR iModulon was initially annotated as malonate

biosynthetic process, but we corrected its function to malonate catabolism as it encodes the malonate catabolic enzyme complex²¹. Finally, iModulon activities across diverse experimental conditions were examined using the built-in graph function of iModulonDB in the individual iModulon page, which can also be obtained by downloading two tables, namely “Experimental Conditions” and “iModulon Activity”. In this step, we took a rational approach to test if changes in iModulon activity under specific conditions were consistent with established biological knowledge. For example, activities of BCAA biosynthetic iModulons and AmpC iModulon are induced in the absence of BCAA and the presence of beta-lactam in the media (**Supplementary Fig. 1C and 2**), affirming the functional annotations.

The strategy for iModulon transfer was established by addressing the genetic context of iModulons. Firstly, gene members of iModulons were extracted by selecting genes with their absolute iModulon weight exceeding the threshold (**Fig. 1B, 2A, 3B, Supplementary Fig. 1A, and 4**) (“Gene Table” in an iModulon page). Next, the genetic organization (location, orientation, and operonic structure) was assessed through Biocyc¹⁸ and Pseudomonas Genome Database⁵³. If genes were found in a single operon, the entire operon was directly amplified and cloned. If genes were located in multiple genomic loci, they were refactored into an operon, while preserving the genetic order to ensure an optimal expression level as demonstrated previously^{25,26}.

Referee 2

Comments for the Authors

This article reports a transcriptomics database tool platform iModulonDB for the holistic heterologous expression of cross-species cell functions. The iModulonDB platform effectively guides expression of various biological functions in different species. The Palsson team demonstrates the extensive applications of this platform in strain design and synthetic biology research through four case studies, including the heterologous expression of *Pseudomonas putida*'s vanillin-to-protocatechuic acid conversion function in *Escherichia coli*, *Pseudomonas aeruginosa*'s malonic acid metabolism function, *Pseudomonas putida*'s 2,3-butanediol metabolism function, and *Pseudomonas aeruginosa*'s ampicillin resistance.

The iModulonDB platform contains a wealth of transcriptomics information, allowing researchers to easily identify genes that may be associated with specific functions of interest in the original host. These genes are overexpressed at the transcriptomics level during the execution of these functions and have been found to be correlated. Some of these genes may have been previously studied in relation to the target function, while others have not received any precedent notice. The work is suitable for the journal.

Suggestions:

1. It is recommended that the control data of the original strain be included in the figures showing cross-species functional restoration experiments. This will allow a more rigorous and intuitive demonstration of the effectiveness of heterologous strains in replicating functions. If relevant data are already available, please directly cite them; otherwise, it is suggested to conduct supplementary experiments.

Response: We acknowledge the reviewer's suggestion regarding the importance of including control data from the original source organism as a reference point for comparing reconstructed functionality. To address this, we have supplemented the original strains' performance in ampicillin resistance (**Fig. 2C, D**), 2,3-BDO degradation (**Fig. 3D**), and malonate utilization (**Fig. 4C**). Notably, the *E. coli* recipients of AmpC and AcoR iModulon exhibited enhanced functionality compared to the original source organism. Specifically, the *E. coli* strain with the full AmpC iModulon demonstrated higher ampicillin resistance (MIC of 4096 µg/ml) than *P. aeruginosa* (MIC of 2048 µg/ml) (**Fig. 2D**) and 2,3-BDO degradation by *E. coli* with AcoR iModulon was faster than that observed in *P. putida*. This enhancement is likely attributed to stronger gene expression in the heterologous host driven by the synthetic inducible promoter (P_{trc}). Nevertheless, iModulon-based engineering proved to be sufficient for the successful reconstruction of cellular functions. Please refer to the revised figures and manuscript as outlined below:

P4, L137-P5, L144: The source of AmpC iModulon, *P. aeruginosa*, showed ampicillin resistance with the minimum inhibitory concentration (MIC) of 2048 µg/ml (**Fig. 2C**). The MIC of ampicillin for laboratory *E. coli* strain MG1655 with empty plasmid was 16 µg/ml, which is comparable to previous reports^{31,32} (**Fig. 2C and 2D**). *E. coli* strain with the *P. aeruginosa* beta-lactamase showed a dramatic increase in ampicillin resistance with an MIC of 1,024 µg/ml, while it was lower than that of the original host (**Fig. 2D**). Strikingly, *E. coli* harboring the entire AmpC iModulon, six auxiliary genes in addition to *ampC*, had an MIC of 4,096 µg/ml, which was four times higher than that with *ampC* alone (**Fig. 2D**).

P6, L190-194: 2,3-BDO dehydrogenase activities of the source organism and *E. coli* strains carrying the three plasmids individually were examined during 96 hours of batch cultivation in LB medium supplemented with 2 g/l of 2,3-BDO. The original strain, *P. putida* KT2440, showed 2,3-BDO utilization with negligible level of acetoin (**Fig. 3D**).

Figure 2. *E. coli* carrying the *Pseudomonas aeruginosa* AmpC iModulon confers better ampicillin resistance than cells expressing beta-lactamase alone. (C) Dose-kill curves of *P. aeruginosa* and *E. coli* carrying empty BAC, BAC_ampC, or BAC_AmpC iM. Error bars indicate s.d. of biological replicates (n=3). Note that the range of ampicillin concentration is different, due to the huge difference in ampicillin tolerance. (D) Cell density of cultures treated with different ampicillin concentrations after 10 hr of incubation. Arrows indicate the minimum inhibitory concentration (MIC). Error bars indicate s.d. of biological replicates (n=3).

Figure 3. Cross-species transfer of 2,3-butanediol (2,3-BDO) utilization iModulon of *Pseudomonas putida* in *E. coli*. (D) 2,3-BDO degradation by *P. putida*. Formation of acetoin was negligible. Dots indicate individual data points. Error bars indicate s.d. of the three biological replicates. (E) 2,3-BDO and acetoin degradation by *E. coli* carrying empty plasmid or one of the three constructs. 2,3-BDO was added at the start of the culture and the remaining amount and acetoin formation was measured. Dots indicate individual data points. Error bars indicate s.d. of the three biological replicates.

In malonate utilization case, in contrast to *E. coli* heterologously expressing MdcR iModulon that consumed malonate very slowly, *P. aeruginosa* was able to utilize malonate as a sole carbon source within 24 hrs of batch cultivation. This illustrates host factors and metabolic changes required to accommodate heterologous iModulons, justifying the pipeline — iModulon transfer followed by ALE. Thanks to the reviewer’s suggestion, we believe that supplementing these control experiments strengthens our manuscript. Please refer to the revised manuscript and figure as follows:

P8, L250-255: *E. coli* could slowly utilize (doubling time of 11.2 ± 0.6 hr; over the course of 72 hours of fermentation in M9 malonate medium) malonate as a carbon source only at weak expression level (**Fig. 4C**). In contrast to complete utilization of malonate by *P. aeruginosa* within 12 hours of fermentation (**Fig. 4C**), the observed slow utilization by *E. coli* suggests a potential metabolic imbalance in *E. coli*, perturbed by and unable to accommodate the malonate pathway.

Figure 4. Adaptive laboratory evolution improves functionality of *Pseudomonas aeruginosa* MdcR iModulon in *E. coli*. (C) Malonate utilization of *P. aeruginosa*, *E. coli* carrying empty plasmid, and pMdcR iM. Cells were incubated for up to 72 hr in M9 malonate (2 g/l) media. Circles and diamonds show cell density and malonate concentration in culture, respectively. Green, gray, and brown lines are *P. aeruginosa*, *E. coli* carrying empty plasmid, and pMdcR iM plasmid, respectively. Error bars indicate s.d. of three replicated cultures.

2. The description of laboratory adaptive evolution (ALE) in this article has flow inconsistency. Moving it to the latter part of the article may better connect with the discussion section.

Response: We appreciate the reviewer’s constructive suggestion. Accordingly, we have reorganized the manuscript to enhance the logical progression of the content. The revised order now features iModulon transfer examples in the sequence of (i) VanR, (ii) AmpC, (iii) AcoR, and (iv) MdcR. We trust this new sequence improves the cohesiveness and the overall readability of the manuscript. Please refer to the revised manuscript and the updates to section order are outlined follows:

RESULTS

1. Cross-species transfer of *Pseudomonas* iModulons into *E. coli*
2. Auxiliary genes may be needed for optimal function of cross-species transferred iModulons
3. Complete iModulon gene membership is needed for successful cross-species transfer
4. ALE optimizes the functionality of catabolic iModulons

Reviewers' Comments:

Reviewer #1:

Remarks to the Author:

The authors have done a great job reorganizing the manuscript and answering all my concerns and I have no further comments at this stage.

Reviewer #2:

Remarks to the Author:

The authors have done thorough work to address my previous concerns. I appreciate the work.